# Unexplained mechanism of subdural hematoma with convulsion suggests nonaccidental head trauma: A multicenter, retrospective study by the Japanese Head injury of Infants and Toddlers study (J-HITs) group

**Ayumi Narisawa[1], Masahiro Nonaka[2]\*, Nobuyuki Akutsu[3], Mihoko Kato[4], Atsuko Harada[5], Young-Soo Park[6]**

1 Department of Neurosurgery, Sendai City Hospital, Miyagi, Japan, 2 Department of Neurosurgery, Kansai Medical University, Osaka, Japan, 3 Department of Neurosurgery, Hyogo Prefectural Kobe Children's Hospital, Hyogo, Japan, 4 Department of Neurosurgery, Aichi Children's Health and Medical Center, Aichi, Japan, 5 Department of Pediatric Neurosurgery, Takatsuki General Hospital, Osaka, Japan, 6 Department of Neurosurgery, Nara Medical University, Nara, Japan

\* nonakamasa65@gmail.com

## Abstract

### Objective

The medical history of injury given by parents of infants and toddlers with head trauma may not be accurate or completely true. The purpose of this study was to examine the relationship between subdural hematoma (SDH) due to nonaccidental injury and mechanisms of injury provided by caregivers.

### Methods

Our multicenter study group retrospectively reviewed the clinical records of children younger than 4 years with head trauma who have been diagnosed with any finding on head computed tomography (CT) and/or magnetic resonance imaging (MRI). A total of 84 cases of subdural hematomas with retinal findings, including cases reported to the child guidance center and traffic and birth injuries, were included in the study. They were classified by the mechanism of injury provided by the caregivers. Clinical findings were reviewed and classified into nonaccidental and accidental groups. The mechanisms of the injuries were examined by multivariable analysis to identify which ones were statistically associated with nonaccidental injuries.

### Results

Of the 84 patients with SDHs, 51 were classified into the nonaccidental group, and 33 children were classified into the accidental group. In 19 patients with a chief complaint of convulsion who had SDH but no episode of trauma, 18 were classified into the nonaccidental

**Data Availability Statement:** All relevant data are within the article and its Supporting Information files.

**Funding:** The authors received no specific funding for this work.

**Competing interests:** I have read the journal's policy and the authors of this manuscript have the following competing interests:Masahiro Nonaka and Young-Soo Park have written statements and appeared in court in child abuse cases both on the request of the prosecutor and the defense. Atsuko Harada has written statements and appeared in court in child abuse cases on the request of the prosecutor. This does not alter our adherence to PLOS ONE policies on sharing data and materials.

group. On multivariable analysis, unexplained convulsions (odds ratio: 12.04, 95% confidence interval: 1.44–100.49) were significantly associated with increased odds of nonaccidental injury.

## Conclusions

In the present study, there was a relationship between nonaccidental injury and unexplained SDH with a chief complaint of convulsion.

## Introduction

As a clinical feature of infants and toddlers, many sources of information in medical interviews are complaints from caregivers. This is also the case for head injuries, and in many cases, caregivers explain when, where, and how the child hit the head. In addition, in many cases, the moment of injury is not witnessed by anyone other than the caregivers at home, and descriptions of the injury situations cannot be obtained from anyone other than the caregivers. On the other hand, although caregivers do not describe episodes of head injuries, convulsions, consciousness disturbance, hemiplegia, etc. may lead to suspicion of abnormalities in the central nervous system, and imaging findings may show head trauma. In the past, unexplained infant head injuries have been thought to be abusive head trauma (AHT) [1–3].

Child Guidance Centers in Japan are administrative agencies established in prefectures and government-designated cities to promote the welfare of children and protect their rights. In cooperation with local communities, Child Guidance Centers provide consultation services to families and others, accurately identify children's problems and environments, and assist children and families. When a child who appears to have been abused is found, it is mandatory to immediately notify the Child Guidance Center.

In the present study, cases of subdural hematoma (SDH) in infants and toddlers younger than 4 years of age were studied [4]. The study included cases of head trauma due to traffic accidents, birth injuries, and other causes that were reported to the Child Guidance Center. They were classified according to the mechanisms of injuries described by their caregivers or what made them go to the hospital, for example, unexplained convulsion or coma. The factors related to accidental or nonaccidental injury were then examined.

## Methods

The clinical records of children younger than 4 years with head trauma who visited hospitals of our study group between January 2014 and August 2020 were retrospectively reviewed. Patients with some imaging findings such as a fracture or intracranial injury were included. The imaging studies examined in this study were either or both of computed tomography (CT) or magnetic resonance imaging (MRI). From the medical charts, the sex and age of the child, mechanism of injury, physical and neurological findings, radiological findings, retinal hemorrhage, surgical intervention, report to child guidance centers, temporary protection by child guidance centers, and criminal cases were extracted. Radiological findings included whether SDH was bilateral or unilateral, and coexistence with brain edema, skull fracture, contusion, acute epidural hematoma (AEDH), subarachnoid hemorrhage (SAH), and arachnoid cyst. Imaging findings were confirmed and recorded by board-certified pediatric neurosurgeons at each institution.

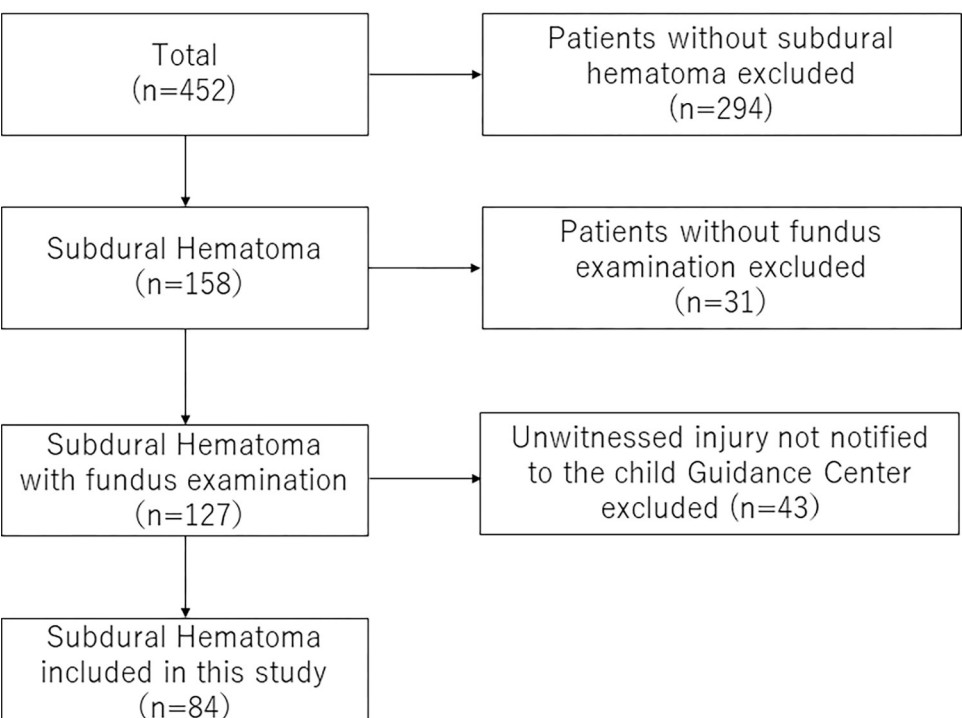

Flow chart of the inclusion and the exclusion criteria.

**Fig 1. Of the total 452 registered cases, 294 without subdural hematoma were excluded.** Of the 158 cases with subdural hematoma, 31 cases without fundus examination were excluded, resulting in 127 cases. In addition, excluding 43 cases that had not been reported to the Child Guidance Center, 84 cases were included in this study.

A total of 452 children were registered, and 158 cases had SDH. Of the cases with SDH, 31 without fundus examinations were excluded. Of the remaining 127 cases, 3 cases were birth injuries, and 4 cases were traffic accidents. Of the other 120 cases, excluding 43 cases that were not reported to the Child Guidance Center, 77 cases, along with the 3 birth injury cases and 4 traffic accident cases, for a total of 84 cases, were included in the present study (Fig 1).

The patients were classified into nonaccidental and accidental groups by our definition, and their clinical presentations were summarized. Nonaccidental injuries included the cases in which the caregivers confessed to abuse, those that were prosecuted without confession of abuse, and those reported to a Child Guidance Center and taken into temporary custody. On the other hand, accidental injuries included the cases that were reported to a Child Guidance Center but not taken into custody, such as traffic accidents and birth injuries (Table 1). Some of the cases in the accident group that was not temporarily protected by the Childs Guidance Center may have been caused by nonaccidental trauma, and some of the cases that were temporarily protected by the Childs Guidance Center may have been caused by accident. However, in Japan, the Childs Guidance Center not only determines the possibility of abuse based on the child's social information, but also conducts a medical evaluation by a third-party

**Table 1. Definitions of "nonaccidental" and "accidental".**

| Nonaccidental | | | Accidental | | |
|---|---|---|---|---|---|
| Perpetrator confesses to abuse. | Perpetrator did not confess to abuse, but was prosecuted | Child guidance center notified, judged as possible abuse, taken into temporary custody | Child guidance center notified, judged as accident, not taken into custody | Traffic accident | Birth injury |

medical expert to more accurately determine whether or not abuse has occurred. Therefore, we consider that only a few cases have been misclassified.

The information about the mechanism of injury was primarily determined from the medical history provided by the caregiver to a physician. The possibility of the disease causing subdural hematoma was determined at each facility. If necessary, medical evaluation reports by child protection teams (CPTs) at each facility and adequate investigation reports by police and/or Child Guidance Center were requested. After the investigation, when caregivers described different mechanisms of injury from the initial medical examination, the final explanation of caregivers was recorded. However, because caregivers do not always state the correct injury mechanism, we classified the case as nonaccidental if the child guidance center decided to take the case into temporary custody, or if the prosecution filed criminal charges as an abuse case. If the opinions differ, the opinion of the investigating agency, such as the police, takes precedence, followed by the opinion of the child guidance center, which takes precedence over the opinion of the CPT of the respective institution.

The injury mechanism described by the caregivers was freely described at each facility. The descriptions were classified into 14 categories in descending order of frequency, and each category was studied. The 14 categories are as follows: birth injury, motor vehicle accident, bicycle accident, fall from the height of over 2 meters, fall with the caregiver holding the patient, drop from being held by the caregiver, self-inflicted fall (from standing or sitting on the flat floor or ground), fall from a bed or sofa (include crib, armrest and backrest of the sofa), other fall from the height of under 2 meters, other head injuries, unexplained convulsions (the chief complaint was convulsion without history of head trauma), unexplained coma (without a history of head trauma), other unexplained events (poor physical condition without a history of head trauma), and confessed abuse.

IRB/ethics committee approval and a statement regarding patient consent.

The protocol for this study was approved by the Ethics Committee of Kansai Medical University (No. 2019232). The need for written patient consent was waived by the Ethics Committee because the data were deidentified. Institutional review board approval was obtained from all participants' institutions before submitting cases for this study.

## Statistical analysis

Statistical analysis was performed with JMP 14.2.0. Univariate and multivariable analyses were performed to examine the relationships between nonaccidental injury and predictors. A univariate logistic model was used to compare each predictor. Variables were included in a multivariable logistic regression model if their *p* value on univariate analysis was significant. The odds ratios (ORs) and 95% confidence intervals (CIs) were calculated.

## Results

Table 2 shows the injury mechanism classification of 84 cases included in the present study.

SDH was found in all 21 cases of unexplained convulsion and in all 7 cases of unexplained coma. Of the 44 cases of self-inflicted falls and 34 cases of falls from a sofa or bed, 48 (61.5%) cases had SDH. The injury mechanism classification and age, sex, clinical symptoms, and imaging findings of the 84 cases are summarized in Table 3.

Of the 84 patients, 51 (60.7%) patients were nonaccidental by our definition; for 12 patients with nonaccidental injury, caregivers confessed to abusing.

Of 30 falls from <2 m (10 of 15 self-inflicted falls, 4 of 5 falls from a bed or sofa, 1 of 5 falls while being held by a parent, and none of the drops from being held by a caregiver), 20 were judged to be accidental injuries by the Child Guidance Center. These patients with SDH due

**Table 2. The injury mechanism classification of 84 cases included in the present study.**

| Cause of injury | | subdural hematoma selected for this study |
|---|---|---|
| Birth injury | | 3 |
| Traffic accidents | Motor vehicle accidents | 3 |
| | Bicycle accidents | 1 |
| Falls from >2m | | 2 |
| Falls from <2m | Falling while being held by a parent | 5 |
| | Parents dropped | 1 |
| | Self-inflicted fall | 15 |
| | Falling from a bed or sofa | 5 |
| | Other falls from <2m | 4 |
| Other head injury | | 5 |
| Unexplained convulsions | | 19 |
| Unexplained coma | | 7 |
| Other unexplained events | | 2 |
| Confessed abuse | | 12 |
| Total | | 84 |

to a self-inflicted fall or fall from a bed or sofa were all age 6 months or older. Twenty-six of the 28 SDH cases (19 unexplained convulsions, 7 unexplained comas, and 2 other unexplained events) in which the caregiver did not state the injury status were considered nonaccidental trauma.

Retinal hemorrhages were documented in 70 of the 84 patients (83.3%). By classification of injury mechanism, retinal hemorrhage was observed in 8 of 12 patients (66.7%) with confessed abuse, in 22 of 26 cases (84.6%) with unexplained convulsion and coma without any episode of trauma, in 15 of 15 cases (100%) with a self-inflicted fall, and in 4 of 5 cases (80%) with a fall from a bed or sofa.

Brain edema existed in 25 of the 84 cases (29.8%). Brain edema was seen in 6 of 12 cases (50%) of confessed abuse. Looking at it in another way, brain edema was observed in 11 of 33 cases (33.3%) considered accidents, and in 14 of 51 cases (27.5%) considered nonaccidents.

On univariate analysis by logistic regression analysis, unexplained convulsions (OR: 17.45, 95% CI: 2.20–138.54), self-inflicted fall (OR: 0.25, 95% CI: 0.076–0.82), under 5 months old (OR: 4.00, 95% CI: 1.58–10.13), and bilateral SDH (OR: 4.33, 95% CI: 1.53–12.26) were significantly associated with increased odds of nonaccidental injury (Table 4).

On multivariable analysis, unexplained convulsion (OR: 12.04, 95% CI: 1.44–100.49) was significantly associated with increased odds of nonaccidental injury (Table 4).

## Discussion

It is difficult to correctly determine whether head injuries of infants and toddlers are accidental or nonaccidental, unless abuse is confessed or witnessed by any adults other than the caregiver. In many cases when infants and toddlers get injured at home, there is no witness other than their caregivers.

Unexplained head injuries of infants and toddlers have been considered AHT. Reece et al reported 287 children with head injuries aged 1 week to 6.5 years [2]. In 56% of those in the definite abuse group, and in 2% of those in the accident group, there was no history to account for the injuries and no history of motor vehicle accidents [2]. Sun et al studied 18 children aged below 5 years of age who presented with subdural hemorrhage without a history of significant trauma [3]. Eleven cases were concluded to be genuine child abuse cases with non-

**Table 3. Injury mechanism classification and detailed profile of 84 cases summarized by age, sex, clinical symptoms, categorization of accidental/nonaccidental, reported to Child Guidance Center, and imaging findings.**

| Cause of injury | | Total | Child guidance center notified | Categorized as nonaccidental trauma | Under 5 months old | Bilateral SDH | Fundus examined | Retinal bleeding | Brain edema | Seizure | Female | Skull fracture | Contusion | AEDH | T-SAH | Arachnoid cyst |
|---|---|---|---|---|---|---|---|---|---|---|---|---|---|---|---|---|
| Birth injury | | 3 | 0 | 0 | 3 | 0 | 3 | 1 | 2 | 0 | 1 | 0 | 0 | 0 | 0 | 0 |
| Traffic accident | Motor vehicle accident | 3 | 0 | 0 | 3 | 0 | 3 | 2 | 1 | 2 | 1 | 1 | 0 | 0 | 0 | 0 |
| | Bicycle accident | 1 | 0 | 0 | 0 | 0 | 1 | 0 | 0 | 0 | 0 | 0 | 0 | 0 | 0 | 1 |
| Fall from >2 m | | 2 | 2 | 0 | 0 | 0 | 2 | 2 | 2 | 0 | 0 | 1 | 0 | 0 | 0 | 0 |
| Fall from <2 m | Fall while being held by a parent | 5 | 5 | 4 | 4 | 3 | 5 | 5 | 0 | 2 | 3 | 3 | 0 | 0 | 1 | 0 |
| | Parents dropped | 1 | 1 | 1 | 1 | 1 | 1 | 1 | 0 | 1 | 1 | 0 | 0 | 0 | 1 | 0 |
| | Self-inflicted fall | 15 | 15 | 5 | 0 | 0 | 15 | 15 | 3 | 9 | 2 | 1 | 0 | 0 | 0 | 0 |
| | Fall from a bed or sofa | 5 | 5 | 1 | 0 | 0 | 5 | 4 | 3 | 2 | 0 | 0 | 0 | 0 | 0 | 0 |
| | Other fall from <2 m | 4 | 4 | 0 | 3 | 3 | 4 | 4 | 2 | 3 | 1 | 1 | 0 | 0 | 1 | 0 |
| Other head injury | | 5 | 5 | 2 | 2 | 2 | 5 | 4 | 0 | 2 | 0 | 1 | 0 | 0 | 1 | 0 |
| Unexplained convulsions | | 19 | 19 | 18 | 14 | 12 | 19 | 15 | 4 | 19 | 10 | 5 | 3 | 1 | 4 | 0 |
| Unexplained coma | | 7 | 7 | 6 | 4 | 3 | 7 | 7 | 2 | 0 | 0 | 1 | 0 | 0 | 1 | 0 |
| Other unexplained events | | 2 | 2 | 2 | 2 | 2 | 2 | 2 | 0 | 0 | 0 | 1 | 1 | 0 | 0 | 0 |
| Confessed abuse | | 12 | 12 | 12 | 9 | 5 | 12 | 8 | 6 | 9 | 4 | 2 | 2 | 0 | 1 | 0 |
| Total | | 84 | 77 | 51 | 45 | 31 | 84 | 70 | 25 | 49 | 23 | 17 | 6 | 1 | 10 | 1 |

Abbreviation used; SDH: subdural hematoma, AEDH: acute epidural hematoma, T-SAH: traumatic subarachnoid hemorrhage

**Table 4. Results of univariate and multivariable analyses to examine the relationship between nonaccidental injury and prognostic factors.**

| cause of the injuty | Univariate analysis | | | | Multivariable analysis | | | |
|---|---|---|---|---|---|---|---|---|
| | odds ratio | p value (Prob>ChiSq) | lower 95% CI | upper 95% CI | odds ratio | p value (Prob>ChiSq) | lower 95% CI | upper 95% CI |
| Under 5 months old | 4.00 | **0.0034** | 1.58 | 10.13 | 2.26 | 0.21 | 0.63 | 8.19 |
| Female | 1.70 | 0.31 | 0.61 | 4.72 | | | | |
| Bilateral SDH | 4.33 | **0.0058** | 1.53 | 12.26 | 1.82 | 0.37 | 0.49 | 6.83 |
| Retinal hemorrhage | 1.19 | 0.76 | 0.37 | 3.82 | | | | |
| Brain edema | 0.76 | 0.567 | 0.29 | 1.96 | | | | |
| Skull fracture | 1.72 | 0.35 | 0.55 | 5.45 | | | | |
| Contusion | N/A | N/A | N/A | N/A | | | | |
| AEDH | N/A | N/A | N/A | N/A | | | | |
| SAH | 6.86 | 0.075 | 0.83 | 56.93 | | | | |
| Arachnoid cyst | N/A | N/A | N/A | N/A | | | | |
| Seizure | 2.40 | 0.056 | 0.98 | 5.90 | | | | |
| Falls from >2m | N/A | N/A | N/A | N/A | | | | |
| Falling while being held by a parent | 2.72 | 0.38 | 0.29 | 25.50 | | | | |
| Parents dropped | N/A | N/A | N/A | N/A | | | | |
| Self-inflicted fall | 0.25 | **0.022** | 0.076 | 0.82 | 1.25 | 0.7593 | 0.31 | 5.08 |
| Other falls from <2m | N/A | N/A | N/A | N/A | | | | |
| Falling from a bed or sofa | 0.15 | 0.091 | 0.015 | 1.36 | | | | |
| Unexplained convulsion | 17.45 | **0.0068** | 2.20 | 138.54 | 12.04 | **0.0215** | 1.44 | 100.49 |
| Unexplained coma | 4.27 | 0.19 | 0.49 | 37.18 | | | | |

Significant p-values are shown in bold. N/A: not available

accidental head injuries [3]. Fung et al reported 7 cases of unexplained SDH [1]. Following multidisciplinary case conferences, a diagnosis of nonaccidental injury was made in 4 cases, and nonaccidental injury was not established in 3 of the 7 cases [1]. In cases with multiple factors such as subdural hematoma, intracranial changes, complicated retinal hemorrhages, and rib and other fractures that are inconsistent with the described injury mechanism, experts state that the medical validity of the presence of AHT is not in dispute [5].

In the present study, multivariable analysis showed a relationship between nonaccidental injury and unexplained SDH with a chief complaint of convulsion. The results of multivariable analysis showed an odds ratio of 12.04, with a 95% CI of 1.44–100.49, for unexplained convulsion. Unexplained here means that the caregiver did not explain or describe the injury mechanism.

However, not all unexplained SDH cases were nonaccidental injuries. In 19 cases that presented with convulsion, 1 case was considered as an accidental injury. In 7 cases that presented with coma, 1 case was considered as an accidental injury. Furthermore, not all mechanisms of injuries in the present study were initially mentioned by caregivers. We checked the consistency of the caregiver's statements, the contradiction of the mechanism of injury and

developmental stage, history of infant medical examination or preventive vaccination, and family background. If necessary, the caregiver was repeatedly asked about the injury situation, and CPTs, Child Guidance Centers, and police investigated whether the injury was assault or not. Then, the mechanism of injury that the caregiver finally stated were finally recorded.

In this study, for non-obvious accidental cases, not only was the decision made by the CPT of the respective hospital, but also by the Child Guidance Center. In Japan, not only does the Child Guidance Center carefully examine the child's social situation to determine whether or not he/she has been abused, but also a third-party medical examiner always appraises the child, and in some cases, the police also conduct a detailed investigation. Although the possibility cannot be ruled out that some of the cases judged to be accidents may have been cases of abuse, as mentioned earlier, they were judged very carefully, and if they existed, they were considered to be a very small number. In the current study, 20 of the 30 cases in which parents stated that SDH was caused by a low fall of 2 m or less were determined to be accidental by the Child Guidance Centers. In Japan, SDH in infants and toddlers has been known to occur due to daily minor head trauma, such as falling backward from standing on a tatami and bruising the back of the head [6, 7]. That is so-called Nakamura's type 1 intracranial hemorrhage, which is reported to be associated with retinal hemorrhage in 60–100% of cases [6, 8, 9].

However, there are many reports that SDHs do not occur due to short-distance falls. Amagasa et al reported that there was no patient with SDH or retinal hemorrhage in all 30 patients younger than 2 years with a fall witnessed by a nonrelative [10]. Lyons et al reported that 207 children younger than 6 years of age who fell from a crib or bed in hospital had no intracranial injury [11].

In the present study, 20 cases corresponded to Nakamura's type 1 intracranial hemorrhage, 15 cases of self-inflicted falls from standing or sitting on a flat floor or ground, and 5 cases of falls from a crib, bed, or sofa. Of these 20 cases, 6 were considered nonaccidental, and retinal hemorrhage was observed in all 6 cases (100%). On the other hand, 14 cases were considered accidental, and retinal hemorrhage was observed in 13 cases (92.9%). All of these 20 patients were over 6 months of age.

## Limitations

Whether the injury mechanism described by the caregiver changed over time, whether a third party witnessed the injury, whether there was a limb and/or rib fracture, and whether there was a bruise, wer)e not investigated. Indications for CT imaging were inconsistent due to factors such as differences between facilities and caregivers' agreement. The timing and indications for fundus examination were not standardized among facilities. Only the presence or absence of fundus hemorrhage was examined, and not the degree and details of the findings. Without considering the timing of convulsions and cerebral edema, only their presence or absence was examined. In this study, only cases that were notified to the Child Guidance Center, which conducted a more multifaceted review of cases of injuries sustained in the home, were included. As a result, the study tended to exclude many minor accident cases that were determined not to require notification, while abused cases tended to be more prevalent. The small number of included cases may limit the power of our statistical analysis.

## Conclusions

In the present study, there was a statistical association between nonaccidental injury and unexplained SDH with a chief complaint of convulsion. The caregivers who are the perpetrators of abuse tended not to explain the mechanism of injury.

## Supporting information

**S1 Appendix. A list of the 84 subdural hematomas studied in this article with respect to age in months, gender, presence of retinal hemorrhage, presence of other intracranial lesions, the parent's stated injury history, and the final administrative determination of abuse.**
(DOCX)

## Author Contributions

**Conceptualization:** Masahiro Nonaka, Young-Soo Park.

**Data curation:** Ayumi Narisawa, Masahiro Nonaka.

**Formal analysis:** Ayumi Narisawa, Masahiro Nonaka.

**Investigation:** Masahiro Nonaka.

**Methodology:** Masahiro Nonaka, Young-Soo Park.

**Project administration:** Masahiro Nonaka.

**Resources:** Ayumi Narisawa, Masahiro Nonaka, Nobuyuki Akutsu, Mihoko Kato, Atsuko Harada, Young-Soo Park.

**Supervision:** Masahiro Nonaka, Young-Soo Park.

**Validation:** Masahiro Nonaka.

**Visualization:** Ayumi Narisawa, Masahiro Nonaka.

**Writing – original draft:** Ayumi Narisawa.

**Writing – review & editing:** Masahiro Nonaka, Nobuyuki Akutsu, Mihoko Kato, Atsuko Harada, Young-Soo Park.

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
