## [Decision Letter · Decision Letter 0]

9 Aug 2022

PONE-D-22-14343Unexplained mechanism of subdural hematoma with convulsion suggests nonaccidental head trauma: A multicenter, retrospective study by the Japanese Head injury of Infants and Toddlers study (J-HITs) groupPLOS ONE

Dear Dr. Nonaka,

Thank you for submitting your manuscript to PLOS ONE. After careful consideration, we feel that it has merit but does not fully meet PLOS ONE’s publication criteria as it currently stands. Therefore, we invite you to submit a revised version of the manuscript that addresses the points raised during the review process.

ACADEMIC EDITOR: I really appreciated the importance of your work. Please address all comments pointed out by the reviewers. Here are some comments from myself.In the abstract and main text, please report odds ratios with 95% CIs. In contrast, p values can be removed.Significant figures should be standardized throughout the manuscript (including tables), e.g., to the second or third decimal place.In general, English-language papers do not use vertical lines in tables. Since there seems to be no compelling reason to do so, please reorganize the tables.Give Figure 1 a title (e.g., Participants flow chart)."multivariate"  "multivariable" Please refer to Hidalgo and Goodman, 2013, Am J Public Health, 103(1), 39-40. DOI: 10.2105/AJPH.2012.300897The use of abbreviations is not good, such as once-defined abbreviations being spelled out again, etc. Please review the manuscript in general.

We look forward to receiving your revised manuscript.

Kind regards,

Kenta Matsumura

Academic Editor

PLOS ONE

Journal Requirements:

Reviewers' comments:

Reviewer's Responses to Questions

**Comments to the Author**

1. Is the manuscript technically sound, and do the data support the conclusions?

Reviewer #1: Partly

2. Has the statistical analysis been performed appropriately and rigorously? 

Reviewer #1: Yes

3. Have the authors made all data underlying the findings in their manuscript fully available?

Reviewer #1: No

4. Is the manuscript presented in an intelligible fashion and written in standard English?

Reviewer #1: No

5. Review Comments to the Author

Reviewer #1: This is a case series from a convenience sample of head injury cases from a larger case registry over 6+ years in Japan. The purpose is to review the presence or absence of a history of how the head trauma occurred and find associations with the mechanism, presentation and determination of accidental versus non-accidental trauma. The finding of the correlation of seizures and a lack of history of the trauma with non-accidental trauma (NAT) is novel and important. However, there are additional statements made regarding the need for a differential diagnosis related to NAT which are not the subject of the research and not included in their methods or results. The references are appropriate given the location of the cases, but there is an additional international reference which would be helpful. In addition, there appear to be issues with the use of English language to be addressed. I have specific comments as follows.

The abstract begins with "true mechanism" where what the authors are trying to say is that the medical history of injury given by parents of head trauma infants may not be accurate or completely true. We do understand the mechanism.

No need to mention the prior study in the abstract. Were there any hypotheses?

There needs to be a better description of the Child Guidance Center and what a report means. Is this what we consider to be a mandated report of suspected abuse to the government for investigation? This should be better explained in the introduction and methods.

The determination of whether the injury is NAT or accidental needs to be better described as this decision is critical to the paper. How were the criteria in Table 1 created? There may be misclassified cases that can easily change the outcome of the study given the small number of cases. You do mention that you take the "last" history provided; how does that bias your results? Perhaps a family has no information and then dreams up a history to match the findings? How do you deal with disagreements among the hospital teams and the investigators and child guidance center? Was there any disagreement?

The abstract talks about the mechanisms increasing the risk of NAT. It is better throughout the paper to refer to these consistently as statistical associations rather than causes or risks.

The abstract also add in a sentence as does the discussion have a paragraph about the differential diagnosis and how they need to be considered. Yet there is no mention whether any of the cases had such an alternative diagnosis and the information is not included in the analysis. While a general statement early on about the important of looking for underlying medical causes is appropriate, such detailed statements with multiple citations are not part of the study and should be deleted.

In the introduction, I would also add the following reference since there is newer information than is discussed:

Choudhary AK, et al. Consensus statement on abusive head trauma in infants and young children. Pediatr Radiol. 2018 Aug;48(8):1048-1065. doi: 10.1007/s00247-018-4149-1.

In methods, I appreciate the description of how the sample was derived from the larger case registry. It would be very helpful to the reader to understand early on what the inclusion criteria were: Under age 4y, SDH on head imaging, reporting to child guidance center, and presence of retinal exam. This severely biases the cases chosen to more serious cases and likely more NAT cases as these are the ones more likely to meet all of these criteria. When cases are not reported to child guidance center, they are likely to be accidental, so this biases your results. Are there exclusion criteria for medical causes? And the small number of cases limits the power of your statistical analyses. This should be listed under limitations.

Also, how were the 14 categories of mechanism determined? There is no explanation in the intro of how these are important. This should be described as well as how the criteria for NAT were decided.

The results section seems appropriate, but why do you discuss all 452 cases? How does that help your findings> You should only focus on the 84.

The conclusions of the paper are that, among children younger than 4 years of age who are reported to the child guidance center, convulsions are significantly associated with NAT in univariate and multivariate models. There is no basis to conclude anything about other medical conditions that were not studied.

6. PLOS authors have the option to publish the peer review history of their article (what does this mean?). If published, this will include your full peer review and any attached files.

Reviewer #1: No

---

## [Author Response · Author response to Decision Letter 0]

6 Sep 2022

Academic Editor

Comment #1 In the abstract and main text, please report odds ratios with 95% CIs. In contrast, p values can be removed.

Author’s Response: The odds ratio with 95% CIs has been added in the abstract and main text according to the academic editor’s suggestions.

Comment # 2 Significant figures should be standardized throughout the manuscript (including tables), e.g., to the second or third decimal place.

Author’s Response: The significant figures has been standardized to the second or third decimal place according to the academic editor’s suggestions.

Comment # 3 In general, English-language papers do not use vertical lines in tables. Since there seems to be no compelling reason to do so, please reorganize the tables.

Author’s Response: The vertical lines in tables　2 and 3 has been removed according to the academic editor’s suggestions.

Comment # 4 Give Figure 1 a title (e.g., Participants flow chart).

Author’s Response: We have given the title in Figure 1.

Comment # 5 "multivariate"  "multivariable" Please refer to Hidalgo and Goodman, 2013, Am J Public Health, 103(1), 39-40. DOI: 10.2105/AJPH.2012.300897

Author’s Response: We have changed the word "multivariate" to "multivariable" according to the editor's suggestion.

Comment # 6 The use of abbreviations is not good, such as once-defined abbreviations being spelled out again, etc. Please review the manuscript in general.

Author’s Response: We apologize for the inappropriate use of abbreviations. We have corrected the use of abbreviations.

Reviewer #1: 

1. Is the manuscript technically sound, and do the data support the conclusions?

Reviewer #1: Partly

Author’s Response: The conclusions have been revised in accordance with the reviewer's comments. The changes are listed below in response to the direct comments.

2. Has the statistical analysis been performed appropriately and rigorously?

Reviewer #1: Yes

3. Have the authors made all data underlying the findings in their manuscript fully available?

Reviewer #1: No

Author’s Response: The data used for this paper has been uploaded as supporting information file (S1).

4. Is the manuscript presented in an intelligible fashion and written in standard English?

Reviewer #1: No

Author’s Response: The English grammar of this paper has been revised

5. Review Comments to the Author

Reviewer #1: 

 This is a case series from a convenience sample of head injury cases from a larger case registry over 6+ years in Japan. The purpose is to review the presence or absence of a history of how the head trauma occurred and find associations with the mechanism, presentation and determination of accidental versus non-accidental trauma. The finding of the correlation of seizures and a lack of history of the trauma with non-accidental trauma (NAT) is novel and important. However, there are additional statements made regarding the need for a differential diagnosis related to NAT which are not the subject of the research and not included in their methods or results. The references are appropriate given the location of the cases, but there is an additional international reference which would be helpful. In addition, there appear to be issues with the use of English language to be addressed. 

I have specific comments as follows.

Comment #1 The abstract begins with "true mechanism" where what the authors are trying to say is that the medical history of injury given by parents of head trauma infants may not be accurate or completely true. We do understand the mechanism.

Author’s Response: Thank you very much for your valuable suggestions. We have made the following changes to the text below.

Change to text：Page 2, Lines 2-3 (with track change)

The medical history of injury given by parents of infants and toddlers with head trauma may not be accurate or completely true.

Comment #2 No need to mention the prior study in the abstract. Were there any hypotheses?

Author’s Response: We prepared this paper in response to a previous study, but as the reviewer pointed out, we have removed the citation in the abstract as we deemed it unnecessary.

Change to text: Page 2, Lines 6-8 (with track change)

Our multicenter study group retrospectively reviewed the clinical records of children younger than 4 years with head trauma who were diagnosed with any finding on head computed tomography (CT) and/or magnetic resonance imaging (MRI).

Comment #3 There needs to be a better description of the Child Guidance Center and what a report means. Is this what we consider to be a mandated report of suspectedabuse to the government for investigation? This should be better explained in the introduction and methods.

Author’s Response: We have added a description of the Child Guidance Centers in Japan and what cases are notified to them.

Change to text: Page 3, Lines 5-10 (with track change)

Child Guidance Centers in Japan are administrative agencies established in prefectures and government-designated cities for the purpose of promoting the welfare of children and protecting their rights. In cooperation with local communities, Child Guidance Centers provide consultation services to families and others, accurately identify children's problems and environments, and assist children and families. When a child who appears to have been abused is found, it is mandatory to immediately notify the Child Guidance Center

Comment #4 (1) The determination of whether the injury is NAT or accidental needs to be better described as this decision is critical to the paper. How were the criteria in Table 1 created? There may be misclassified cases that can easily change the outcome of the study given the small number of cases. 

Author’s Response: Table 1 reflects the results of separating abuse and accidents as much as possible. It is possible that some of the cases in the accident group that were not temporarily protected by the e Childs Guidance Center may have been caused by NAT, and some of the cases that were temporarily protected by the Childs Guidance Center may have been caused by accident. However, in Japan, the Childs Guidance Center not only determines the possibility of abuse based on the child's social information, but also conducts a medical evaluation by a third-party medical expert to more accurately determine whether or not abuse has occurred. We believe that we have achieved a certain level of accuracy. Therefore, we consider that this classification has the lowest margin of error at this time.

Change to text: Page 4, Lines 12-19 (with track change)

Some of the cases in the accident group that were not temporarily protected by the Childs Guidance Center may have been caused by nonaccidental trauma, and some of the cases that were temporarily protected by the Childs Guidance Center may have been caused by accident. However, in Japan, the Childs Guidance Center not only determines the possibility of abuse based on the child's social information, but also conducts a medical evaluation by a third-party medical expert to more accurately determine whether or not abuse has occurred. Therefore, we consider that only a few cases have been misclassified.

Comment #4 (2) You do mention that you take the "last" history provided; how does that bias your results? Perhaps a family has no information and then dreams up a history to match the findings? 

Author’s Response: The injury mechanism described by the caregivers was carefully reviewed in the CPT at each facility. If there was any doubt, the case was notified to the Child Guidance Center. If the Child Guidance Center decided to take temporary custody of the case, or if the prosecution filed a criminal complaint as an abuse case, the case was classified as nonaccidental in this study. This allowed us to examine how many cases of nonaccidental injuries exist in the caregivers' last stated injury scenario.

Change to text: Page 5, Lines 7-10. (with track change)

However, because caregivers do not always state the correct injury mechanism, we classified the case as nonaccidental if the child guidance center decided to take the case into temporary custody, or if the prosecution filed criminal charges as an abuse case. 

Comment #4 (3) How do you deal with disagreements among the hospital teams and the investigators and child guidance center? Was there any disagreement?

Author’s Response: If there is a difference of opinion, the opinion of the police or other investigating agency takes precedence, followed by the opinion of the Child Guidance Center, which takes precedence over the opinion of the CPT of the respective institution. However, even if the prosecutor does not prosecute, the Child Guidance Center may take custody of the child if abuse is suspected.

Change to Text: Page 5, Lines 10-14 (with track change)

If the opinions differ, the opinion of the investigating agency, such as the police, takes precedence, followed by the opinion of the child guidance center, which takes precedence over the opinion of the CPT of the respective institution.

Comment #5 The abstract talks about the mechanisms increasing the risk of NAT. It is better throughout the paper to refer to these consistently as statistical associations rather than causes or risks. 

Author’s Response: We have changed the term “risk factor” to “statistical association”

Change to text : Page 2, Lines 13-14 (with track change)

The mechanisms of the injuries were examined by multivariable analysis to identify which ones were statistically associated with nonaccidental injuries.

Comment #6 The abstract also add in a sentence as does the discussion have a paragraph about the differential diagnosis and how they need to be considered. Yet there is no mention whether any of the cases had such an alternative diagnosis and the information is not included in the analysis. While a general statement early on about the important of looking for underlying medical causes is appropriate, such detailed statements with multiple citations are not part of the study and should be deleted.

Author’s Response: In the abstract, we deleted the description of differential diagnosis of medical causes which was not addressed in this study.

Change to text; Page 2, Lines 22. (with track change)

The sentence “However, in patients with an unexplained injury mechanism, it is necessary to make a careful differential diagnosis from intrinsic diseases” was removed.

Comment #7 In the introduction, I would also add the following reference since there is newer information than is discussed:

Choudhary AK, et al. Consensus statement on abusive head trauma in infants and young children. Pediatr Radiol. 2018 Aug;48(8):1048-1065. doi: 10.1007/s00247-018-4149-1.

Author’s Response: We cited the literature as suggested by the reviewers and discussed its contents.

Change to text : Page 9, Lines 18-21. (with track change)

 In cases with multiple factors such as subdural hematoma, intracranial changes, complicated retinal hemorrhages, and rib and other fractures that are inconsistent with the described injury mechanism, experts state that the medical validity of the presence of AHT is not in dispute[4].

Comment #8 (1) In methods, I appreciate the description of how the sample was derived from the larger case registry. It would be very helpful to the reader to understand early on what the inclusion criteria were: Under age 4y, SDH on head imaging, reporting to child guidance center, and presence of retinal exam. This severely biases the cases chosen to more serious cases and likely more NAT cases as these are the ones more likely to meet all of these criteria. When cases are not reported to child guidance center, they are likely to be accidental, so this biases your results. 

Author’s Response: Many cases were not notified to the Child Guidance Center because of the high possibility of accidents, but it is possible that the CPT at each facility does not have a firm determination of abuse or accident. For this reason, we did not include cases that were not reported to the Child Guidance Center in this study. 

Change to Text: Page 11, Lines 16-21. (with track change)

 In this study, only cases that were notified to the Child Guidance Center, which conducted a more multifaceted review of cases of injuries sustained in the home, were included. As a result, the study tended to exclude many minor accident cases that were determined not to require notification, while abused cases tended to be more prevalent.

Comment #8 (2) Are there exclusion criteria for medical causes?

Author’s Response: There are no common medical exclusion criteria across facilities. However, it is common practice in Japan to check family history and exclude hemorrhagic disease or skeletal dysplasia when a case of abuse is suspected.

Change to Text: Page 5, Lines 2-3. (with track change)

The possibility of the disease causing subdural hematoma was determined at each facility. 

Comment #8 (3) And the small number of cases limits the power of your statistical analyses. This should be listed under limitations.

Author’s Response: The number of cases included in this report limits the power of statistical analysis.

Change to text: Page 11, Lines 20-21. (with track change)

 The small number of included cases may limit the power of our statistical analysis.

Comment #9 Also, how were the 14 categories of mechanism determined? There is no explanation in the intro of how these are important. This should be described as well as how the criteria for NAT were decided.

Author’s Response: In this study, we asked each facility to freely describe the sequence of events that led to the injury. After reviewing the descriptions, we found that they can be categorized into 14 items.

Change to text: Page 5, Lines 14-16. (with track change)

The injury mechanism described by the caregivers was freely described at each facility. The descriptions were classified into 14 categories in descending order of frequency, and each category was studied.

Comment #10 The results section seems appropriate, but why do you discuss all 452 cases? How does that help your findings> You should only focus on the 84.

Author’s Response: To consolidate to 84 cases, we removed the number of causes of trauma for all 452 enrolled cases listed in Table 2

Change to Text: We have removed data on all 452 cases from Table 2.

Comment #11 The conclusions of the paper are that, among children younger than 4 years of age who are reported to the child guidance center, convulsions are significantly associated with NAT in univariate and multivariate models. There is no basis to conclude anything about other medical conditions that were not studied.

Author’s Response: The description about medical condition in conclusions has been deleted.

Change to text: Page 11, Lines 25. (with track change)

 The sentence “However, in patients with an unexplained injury mechanism, if head trauma is observed on imaging, it is necessary to carefully make a differential diagnosis from intrinsic diseases that present with findings similar to those of trauma. “ has been deleted

---

## [Decision Letter · Decision Letter 1]

11 Oct 2022

PONE-D-22-14343R1Unexplained mechanism of subdural hematoma with convulsion suggests nonaccidental head trauma: A multicenter, retrospective study by the Japanese Head injury of Infants and Toddlers study (J-HITs) groupPLOS ONE

Dear Dr. Nonaka,

Thank you for submitting your manuscript to PLOS ONE. After careful consideration, we feel that it has merit but does not fully meet PLOS ONE’s publication criteria as it currently stands. Therefore, we invite you to submit a revised version of the manuscript that addresses the points raised during the review process.

We look forward to receiving your revised manuscript.

Kind regards,

Kenta Matsumura

Academic Editor

PLOS ONE

Journal Requirements:

Additional Editor Comments:

Thank you for the revised manuscript. The reviewers are generally favorable to the revised manuscript, but there are still some minor corrections to be made. Because PLoS One does not edit manuscripts after acceptance, any spelling errors should be corrected by this time. We strongly recommend that authors use an editing service or have their manuscript checked by a native speaker with expertise in the field. If possible, please submit the revised manuscript with a certificate of proofreading by a third party.

Reviewers' comments:

Reviewer's Responses to Questions

**Comments to the Author**

1. If the authors have adequately addressed your comments raised in a previous round of review and you feel that this manuscript is now acceptable for publication, you may indicate that here to bypass the “Comments to the Author” section, enter your conflict of interest statement in the “Confidential to Editor” section, and submit your "Accept" recommendation.

Reviewer #1: All comments have been addressed

2. Is the manuscript technically sound, and do the data support the conclusions?

Reviewer #1: Yes

3. Has the statistical analysis been performed appropriately and rigorously? 

Reviewer #1: Yes

4. Have the authors made all data underlying the findings in their manuscript fully available?

Reviewer #1: Yes

5. Is the manuscript presented in an intelligible fashion and written in standard English?

Reviewer #1: Yes

6. Review Comments to the Author

Reviewer #1: Thank you for addressing my concerns. The paper is substantially better, but still has some minor corrections needed:

p5. add space between the and disease at the end of line 2.

Add a period at the end of the paragraph "and a statement regarding patient consent."

p6. In table 2, please align the cause of injury to the left to aid readability.

p7. In table 3, please increase spacing in the heading so letters are not dropped to the next line.

p8. Please align all paragraphs to the left.

p9. Please add a space in the second paragraph of the discussion between accidents and (10).

p10.Change second line to say "...inconsistent with the described injury mechanism, there is consensus that the medical validity..."

p11. remove space at end of sentence in line 3.

Correct spelling to "were" in the line 3 of the third paragraph.

7. PLOS authors have the option to publish the peer review history of their article (what does this mean?). If published, this will include your full peer review and any attached files.

Reviewer #1: No

---

## [Author Response · Author response to Decision Letter 1]

15 Oct 2022

Reviewer #1: 

Comment #1 p5. add space between the and disease at the end of line 2.

Author’s Response: The text has been revised according to the reviewer`s suggestion.

Comment #2 Add a period at the end of the paragraph "and a statement regarding patient consent."

 Author’s Response: The text has been revised according to the reviewer`s suggestion.

Comment #3 p6. In table 2, please align the cause of injury to the left to aid readability.

Author’s Response: The table has been revised according to the reviewer`s suggestion.

Comment #4 p7. In table 3, please increase spacing in the heading so letters are not dropped to the next line.

Author’s Response: The table has been revised according to the reviewer`s suggestion.

Comment #5 p8. Please align all paragraphs to the left.

Author’s Response: The paragraphs has been revised according to the reviewer`s suggestion.

Comment #6 p9. Please add a space in the second paragraph of the discussion between accidents and (10).

Author’s Response: The paragraphs has been revised according to the reviewer`s suggestion.

Comment #7 p10.Change second line to say "...inconsistent with the described injury mechanism, there is consensus that the medical validity..."

Author’s Response: The text has been revised according to the reviewer`s suggestion.

Comment #8 p11. remove space at end of sentence in line 3.

Author’s Response: The text has been revised according to the reviewer`s suggestion.

Comment #9 Correct spelling to "were" in the line 3 of the third paragraph.

Author’s Response: The text has been revised according to the reviewer`s suggestion.

---

## [Editor Report · Decision Letter 2]

20 Oct 2022

Unexplained mechanism of subdural hematoma with convulsion suggests nonaccidental head trauma: A multicenter, retrospective study by the Japanese Head injury of Infants and Toddlers study (J-HITs) group

PONE-D-22-14343R2

Dear Dr. Nonaka,

We’re pleased to inform you that your manuscript has been judged scientifically suitable for publication and will be formally accepted for publication once it meets all outstanding technical requirements.

Kind regards,

Kenta Matsumura

Academic Editor

PLOS ONE
---

## [Editor Report · Acceptance letter]

24 Oct 2022

PONE-D-22-14343R2 

Unexplained mechanism of subdural hematoma with convulsion suggests nonaccidental head trauma: A multicenter, retrospective study by the Japanese Head injury of Infants and Toddlers study (J-HITs) group 

Dear Dr. Nonaka:

I'm pleased to inform you that your manuscript has been deemed suitable for publication in PLOS ONE. Congratulations! Your manuscript is now with our production department. 

Kind regards, 

on behalf of

Dr. Kenta Matsumura 

Academic Editor

PLOS ONE